# Contrastive Hierarchical Clustering

## Abstract

Deep clustering has been dominated by flat clustering models, which split a dataset into a predefined number of groups. Although recent methods achieve extremely high similarity with the ground truth on popular benchmarks, the information contained in the flat partition is limited. In this paper, we introduce CoHiClust, a Contrastive Hierarchical Clustering model based on deep neural networks, which can be applied to large-scale image data. By employing a self-supervised learning approach, CoHiClust distills the base network into a binary tree without access to any labeled data. The hierarchical clustering structure can be used to analyze the relationship between clusters as well as to measure the similarity between data points. Experiments performed on typical image benchmarks demonstrate that CoHiClust generates a reasonable structure of clusters, which is consistent with our intuition and image semantics. Moreover, by applying the proposed pruning strategy, we can restrict the hierarchy to the requested number of clusters (leaf nodes) and obtain the clustering accuracy outperforming existing hierarchical baselines.

## 1 Introduction

Clustering, a fundamental branch of unsupervised learning, is often one of the first steps in data analysis, which finds applications in anomaly detection Barai & Dey (2017), personalized recommendations Zhang et al. (2014) or bioinformatics Lakhani et al. (2015). Since it does not use any information about class labels, representation learning becomes an integral part of deep clustering methods. Initial approaches use representations taken from pre-trained models Guérin et al. (2017); Naumov et al. (2021) or employ auto-encoders in joint training of the representation and the clustering model Guo et al. (2017a); Mautz et al. (2019). More recent works designed to image data frequently follow the self-supervised learning principle, where representation is trained on pairs of similar images automatically generated by data augmentations Li et al. (2021b); Dang et al. (2021). Since augmentations used for image data are class invariant, the latter techniques often obtain very high similarity with the ground truth classes. However, we should be careful when comparing clustering techniques only by inspecting their accuracy with ground truth classes because the primary goal of clustering is to deliver information about data and not to perform classification.

Most works in the area of deep clustering focus on producing flat partitions with a predefined number of groups. Although hierarchical clustering gained notable attention in classical machine learning and has been frequently applied in real-life problems Zou et al. (2020); Śmieja et al. (2014), its role has been drastically marginalized in the era of deep learning. In the case of hierarchical clustering, the exact number of clusters does not have to be specified because we can inspect the partition at various tree levels. Moreover, we can analyze the clusters' relationships, e.g. by finding superclusters or measuring the distance between groups in the hierarchy. The above advantages make hierarchical clustering an excellent tool for analyzing complex data. However, in order to take full advantage of hierarchical clustering, it is necessary to create an appropriate image representation, which is possible thanks to the use of deep neural networks. To the best of our knowledge, DeepECT Mautz et al. (2019; 2020) is the only hierarchical clustering model trained jointly with the neural network. Nevertheless, this method has not been examined to large image datasets, which appear in practical applications.

To fill this gap, we introduce CoHiClust (**Co**ntrastive **Hi**erarchical **Clust**ering), which creates a hierarchy of clusters and can be applied to large image data. CoHiClust uses a neural network to generate a high-level representation of data, which is next distilled into the tree hierarchy by applying the projection head, see Figure 2. The whole framework is trained jointly in an end-

to-end manner without labels using our novel contrastive loss and automatically generated data augmentations following the self-supervised learning principle.

The constructed hierarchy uses the structure of a binary tree, where the sequence of decisions made by internal nodes determines the final assignment to clusters (leaf nodes). In consequence, similar examples are processed longer by the same path than dissimilar ones. By inspecting the number of edges needed to connect two clusters (leaves), we obtain a natural similarity measure between data points. Although CoHiClust assumes a pre-defined tree structure with a fixed height, we introduce a pruning mechanism, which removes the least informative leaf nodes until the requested number of leaves is obtained. In contrast to typical pruning strategies or tree cuts, where neighboring leaves are only merged to ancestor node, we allow for reassigning data points between clusters by finetuning the whole model, which further improves topology of the tree.

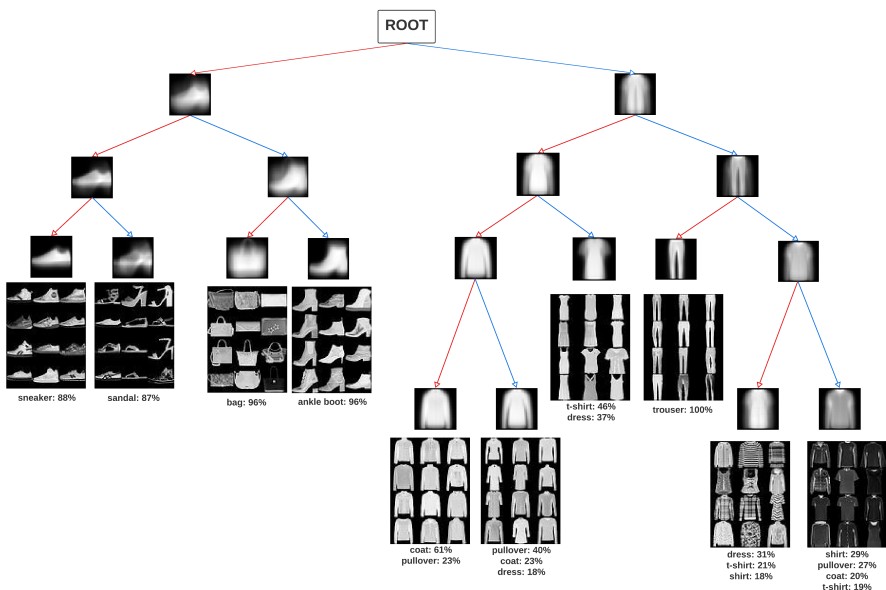

Figure 1: **A tree hierarchy generated by CoHiClust for F-MNIST** (images in the nodes denote mean images in each sub-tree). The right sub-tree contains clothes while the other items (shoes and bags) are placed in the left branch. Looking at the lowest hierarchy level, we have clothes with long sleeves grouped in the neighboring leaves. The same holds for clothes with designs. Observe that CoHiClust assigned white-colored t-shirts and dresses to the same cluster, while trousers are in the separate one. Small shoes such as sneakers or sandals are considered similar (neighboring leaves) and distinct from ankle shoes. Concluding, CoHiClust is able to retrieve meaningful information about image semantics, which is complementary to the ground truth classification.

The proposed model has been examined on various image datasets and compared with both hierarchical and flat clustering baselines. By analyzing the constructed hierarchies, we show that CoHiClust generates a structure of clusters that is consistent with our intuition and image semantics, see Figures 1 for the illustration and discussion. Our analysis is supported by a quantitative study, which shows that CoHiClust gives higher similarity with ground truth partition than available hierarchical baselines, see Tables 1, 2 and 3. Moreover, it is among the three best-performing methods when compared to the flat clustering models, see Table 3.

Our main contributions are summarized as follows:

- We introduce a hierarchical clustering model CoHiClust, which converts the base neural network into a binary tree. The model is trained effectively with no supervision using our novel hierarchical contrastive loss applied to self-generated data augmentations.
- We implement a pruning strategy, which not only leads to creating a fixed number of leaves but also improves the constructed hierarchy.

- Our experimental analysis shows that CoHiClust builds hierarchies based on the well-defined and intuitive patterns retrieved from data.
- Since CoHiClust is the first deep hierarchical clustering model applied to large-scale image datasets, we also deliver a new benchmark, which can be used to compare hierarchical clustering methods.

## 2 RELATED WORK

In this section, we briefly introduce some recent developments in three related topics, i.e., contrastive learning, deep clustering, and hierarchical methods.

**Contrastive Learning**   The basic idea of contrastive learning is to learn such a feature space, in which similar pairs stay close to each other while dissimilar ones are far apart Chopra et al. (2005). In recent works, it was observed that in selected domains, such as computer vision, positive (similar) pairs can be generated automatically using adversarial perturbations Miyato et al. (2018) or data augmentation He et al. (2020), giving the rise of a new field called self-supervised learning Chen et al. (2020). Fine-tuning a simple classifier on self-supervised representation allows for obtaining the accuracy comparable to a fully supervised setting. SimCLR He et al. (2020) applies NT-Xent loss to maximize the agreement between differently augmented views of the same sample. Barlow Twins Zbontar et al. (2021) learns to make the cross-correlation matrix between two distorted versions of the same samples close to the identity. BYOL Grill et al. (2020) claims to achieve new state-of-the-art results without using negative samples. Other works use memory banks to reduce the cost of computing the embeddings of negative samples in every batch Wu et al. (2018); He et al. (2020).

**Deep clustering**   A primary focus in deep embedded clustering has merely been on flat clustering objectives with the actual number of clusters known a priori. DEC Xie et al. (2016) is one of the first works, which combines the auto-encoder loss with a clustering objective to jointly learn the representation and perform clustering. This idea was further explored with some improvements in IDEC Guo et al. (2017a), JULE Yang et al. (2016) and DCEC Guo et al. (2017b). IMSAT Hu et al. (2017) and IIC Ji et al. (2019) use perturbations to generate pairs of similar examples and apply information-theoretic objectives for training. PICA Huang et al. (2020) maximizes the global partition confidence of the clustering solution to find the most semantically plausible data separation. Following the progress in self-supervised learning, CC Li et al. (2021b) and DCSC Zhang et al. (2022) perform contrastive learning by generating pairs of positive and negative instances through data augmentations.

**Hierarchical methods**   Hierarchical clustering algorithms are a well-established area within classical data mining Murtagh & Contreras (2012), but they were rarely studied in deep learning. Deep-ECT Mautz et al. (2019; 2020) is the only method, which jointly learns the deep representation using auto-encoder architecture and performs hierarchical clustering in a top-down manner. Unfortunately, there is not comparative study conducted on large image data. The experimental study of objective-based hierarchical clustering methods performed on the embedding vectors from pre-trained deep learning models is presented in Naumov et al. (2021). In the case of classification, there is a growing interest in deep hierarchical methods, which in our opinion should also be reflected in the area of unsupervised learning. SDT Frosst & Hinton (2017) is one of the first models that distills the base neural networks into a soft decision tree. More advanced methods automatically generate deep networks with a tree structure in a multi-step or an end-to-end manner Tanno et al. (2019); Alaniz et al. (2021); Wan et al. (2020).

## 3 COHICLUST MODEL

The proposed CoHiClust builds a hierarchy of clusters based on the output of the base neural network. There are four key components of CoHiClust:

- The base neural network, which generates representation used by the tree.
- The tree model, which assigns data points to clusters by a sequence of decisions.
- The regularized contrastive loss, which allows for training the whole framework.

- The pruning strategy, which restricts the tree to the requested number of leaves.

We discuss the above components in detail in the following parts.

**Tree hierarchy**   We use a soft binary decision tree to create a hierarchical structure, where leaves play the role of clusters (similar to Frosst & Hinton (2017)). In contrast to hard decision trees, every internal node defines the probability of taking a left/right branch. The final assignment of the input examples to clusters involves partial decisions made by the internal nodes. Aggregating these decisions induces the posterior probability over leaves.

Let us consider a complete binary tree with $T$ levels, where the root is located at the level $0$ and the leaves are represented at the level $T$. This gives us $2^t$ nodes at the level $t$ denoted by tuples $(t, i)$, for $i = 0, 1, \ldots, 2^t - 1$, see Figure 2. The path going from the root to the node $(t, i)$ is given by the sequence of binary decisions $y = (y_1, \ldots, y_t) \in \{0, 1\}^t$ made by the internal nodes, where $y_s = 0$ ($y_s = 1$) means that we take the left (right) branch being in the node at the level $s$. Observe that we can retrieve the index $j$ of the node at the level $s$ from $y$ by taking $j = b_s(y) = \sum_{m=1}^{s} y_m 2^{s-m}$. In other words, the first $s$ bits of $y$ are a binary representation of the number $j$.

We consider the path induced by the sequence of decisions $y = (y_1, \ldots, y_t) \in \{0, 1\}^t$, which goes from the root to the node $(t, i)$, where $i = b_t(y)$. We want to calculate the probability $P_t^i(x)$ that the input example $x \in \mathbb{R}^D$ reaches node $(t, i)$. If $p_s^{b_s(y)}(x)$ is the probability of going from the parent node $(s-1, b_{s-1}(y))$ to its descendant $(s, b_s(y))$, then

$$P_t^i(x) = p_1^{b_1(y)}(x) \cdot p_2^{b_2(y)}(x) \cdot \ldots \cdot p_t^{b_t(y)}(x).$$

Observe that $P_t(x) = [P_t^0(x), P_t^1(x), \ldots, P_t^{2^t - 1}(x)]$ defines a proper probability distribution, i.e. $\sum_{j=0}^{2^t - 1} P_t^j(x) = 1$. In consequence, the probability distribution over clusters (leaves) equals $P_T(x)$, see Figure 2.

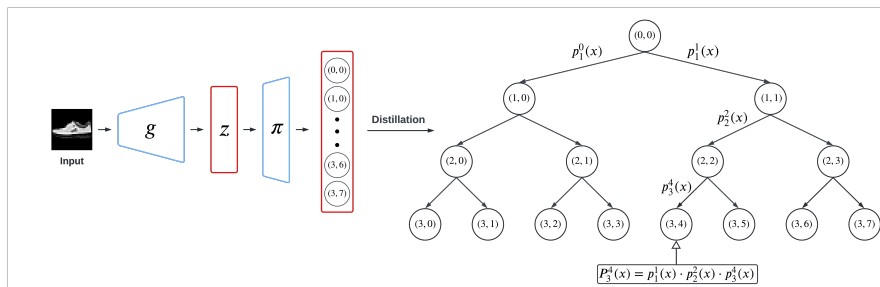

Figure 2: **Illustration of CoHiClust.** The output neurons of the projection head $\pi$ (appended to the base network $g$) model decisions made by the internal tree nodes. The final assignment of the input example to the cluster (leaf node) is performed by aggregating edge probabilities located on the path from the root to this leaf.

**Tree generation**   To generate our tree model, we need to parameterize the probabilities $p_t^i(x)$ of taking the left/right branch in every internal node. To this end, we employ a neural network $g : \mathbb{R}^D \to \mathbb{R}^N$ with an additional projection head $\pi : \mathbb{R}^N \to \mathbb{R}^K$, where $K = 2^T - 1$ and $T$ is the height of the tree. The number $K$ of the output neurons equals the number of internal tree nodes.

The neural network $g$ is responsible for extracting high-level information from data. It can be instantiated by a typical architecture, such as ResNet, and is used to generate embeddings $z = g(x)$ of the input data. We do not use pre-trained networks but train the whole model end-to-end using the proposed hierarchical contrastive loss. This allows for generating the representation, which is suited to the underlying clustering task.

The projection head $\pi$ operates on the embeddings $z$ and parameterizes decisions made by the internal tree nodes. In our case, $\pi$ is a single-layer network with the output dimension equal to the number of internal nodes. To model binary decisions of the internal nodes, we apply the sigmoid function $\sigma$. In consequence, the projection head is given by $\pi(z) = [\sigma(w_1^T z + b_1), \ldots, \sigma(w_K^T z + b_K)]$, where

$w_n \in \mathbb{R}^N$ and $b_n \in \mathbb{R}$ are trainable parameters of $\pi$. By interpreting the output neurons of $\pi$ as the internal nodes of the decision tree, we get the probabilities of left edges in the nodes:

$$p_{t+1}^{2i}(x) = \sigma(w_n^T z + b_n) \text{, for } n = 2^t + i.$$

Note that $p_{t+1}^{2i-1}(x) = 1 - p_{t+1}^{2i}(x)$ always corresponds to the probability of right edge.

**Contrastive hierarchical loss**   To train CoHiClust, we introduce the hierarchical contrastive loss function designed for trees. Our idea relies on maximizing the likelihood that similar data points will follow the same path. The more similar data points, the longer they should be routed through the same nodes. Since we work in the unsupervised setting, we use a self-supervised approach and generate similar images by data augmentations.

Let us consider two data points $x_1, x_2$ and their posterior probabilities $P_t(x_1), P_t(x_2)$ at the level $t$. The probability that $x_1$ and $x_2$ reach the same node at this level is given by the scalar product $P_t(x_1) \cdot P_t(x_2) = \sum_{i=0}^{2^t-1} P_t^i(x_1) P_t^i(x_2)$. This term is maximal if both probabilities are identical one-hot vectors. In a training phase, we do not want to force hard splits in the nodes (binary probabilities) because in this way the model quickly finds a local minimum by assigning data points to fixed leaves with high confidence. Instead of sticking to hard assignments in a few training epochs, we want to let the model explore possible solutions. To this end, we take the square root before applying the scalar product, which corresponds to the Bhattacharyya coefficient Bhattacharyya (1946):

$$s_t(x_1, x_2) = \sqrt{P_t(x_1) \cdot P_t(x_2)} = \sum_{i=0}^{2^t-1} \sqrt{P_t^i(x_1) P_t^i(x_2)}. \tag{1}$$

Observe that $s_t(x_1, x_2) = 1$, if only $P_t(x_1) = P_t(x_2)$ (probabilities do not have to binarize), which leads to the exploration of possible paths. By aggregating the similarity scores over all tree levels, we arrive at our final similarity function $s(x_1, x_2) = \sum_{t=0}^{T-1} s_t(x_1, x_2)$.

In a training phase, we take a minibatch $\{x_j\}_{j=1}^N$ of $N$ examples and generate its augmented view $\{\tilde{x}_j\}_{j=1}^N$. Every pair $(x_j, \tilde{x}_j)$ is considered positive, which means that we will maximize their similarity score. In consequence, we encourage the model to assign them to the same leaf node. To avoid degenerate solutions, where all points end up in the same leaf, we treat all other pairs negative and minimize the similarity scores for them. Finally, the proposed hierarchical contrastive loss is given by:

$$\text{CoHiLoss} = \frac{1}{N(N-1)} \sum_{j=1}^N \sum_{i \neq j} s(x_j, \tilde{x}_i) - \frac{1}{N} \sum_{j=1}^N s(x_j, \tilde{x}_j).$$

By minimizing the above loss, we maximize the likelihood that similar data points follow the same path (second term) and minimize the likelihood that dissimilar ones are grouped together.

**Regularization**   Final cluster assignments are induced by aggregating several binary decisions made by the internal tree nodes. In practice, the base neural network may not train all nodes and, in consequence, use only a few leaves for clustering. While selecting the number of clusters in flat clustering is desirable, here we would like to create a hierarchy, which is not restricted to a given number of leaves. To enable end-to-end training of the base neural network with the arbitrary number of leaves, we consider two regularization strategies.

The first regularization (dubbed $R_1$) explicitly encourages the model to use both left and right sub-trees equally Frosst & Hinton (2017). We realize this postulate by minimizing the cross-entropy between the desired distribution $[0.5, 0.5]$ and the actual distribution for choosing the left or right path in a given node.

The second regularization (dubbed $R_2$) does not directly influence the routing in the tree but focuses on improving the output representation of the base network $g$. For this purpose, we use NT-Xent loss Chen et al. (2020) on the embedding space $z = g(x)$. With the NT-Xent loss, we maximize the cosine similarity on the embedding space for all positive pairs and minimize the cosine similarity on the embedding space for all negative pairs.

Taking together the contrastive loss CoHiLoss with two regularization functions $R_1$ (for entropy) and $R_2$ (for NT-Xent), we arrive at our final loss:

$$\text{Loss} = \text{CoHiLoss} + \beta_1 R_1 + \beta_2 R_2, \tag{2}$$

where $\beta_1, \beta_2$ are the hyperparameters defining the importance of regularization terms $R_1$ and $R_2$, respectively. To generate a full hierarchy (complete tree with the assumed height), we set $\beta_1$ proportional to the depth of the tree $\beta_1 = 2^{-T}$ and $\beta_2 = 1$. In appendix, we show that CoHiClust can also detect the number of clusters automatically by putting $\beta_1 = 0$ (see appendix for the ablation study).

**Pruning**    The proposed model builds a complete tree with $2^T$ leaves. Although such a structure is useful for analyzing the hierarchy of clusters, in some cases we are interested in creating a tree with the requested number of groups. For this purpose, we introduce a pruning strategy that reduces the least significant leaf nodes.

We start with calculating the probability of leaves $P_T^i = \frac{1}{|X|} \sum_{x \in X} P_T^i(x)$, which describes the expected fraction of data points assigned to a given leaf. Assuming that the importance of the cluster is related to the average number of assigned examples, we reduce a leaf with the lowest probability. After removing the leaf, we fine-tune the whole model using CoHiClust loss (2). If we want to reduce more leaves, we perform leaf pruning and model retraining iteratively until the requested number of leaves is obtained. The resulting tree is binary, but can have an arbitrary structure and does not have to be complete. Consequently, pruning is also a way of improving the topology of the tree model.

Alternatively, we could also perform the pruning strategy without applying fine-tuning step. This potentially saves the computational resources, but does not adjust hierarchy to the modified number of groups. In appendix, we show that this approach generates partition of lower quality than the model with fine-tuning.

## 4    EXPERIMENTS

Table 1: Dendrogram purity (DP) of clustering hierarchies (higher is better).

| Method | MNIST | F-MNIST |
|---|---|---|
| DeepECT | 0.82 | 0.47 |
| DeepECT + Aug | 0.94 | 0.44 |
| IDEC + Single | 0.39 | 0.34 |
| IDEC + Complete | 0.40 | 0.35 |
| AE + Bisecting | 0.53 | 0.38 |
| AE + Single | 0.11 | 0.10 |
| AE + Complete | 0.25 | 0.26 |
| **CoHiClust** | 0.97 | 0.52 |

Table 2: Clustering metrics of partitions generated from the tree hierarchies on MNIST and F-MNIST (higher is better).

| Method | MNIST | | F-MNIST | |
|---|---|---|---|---|
| | NMI | ACC | NMI | ACC |
| DeepECT | 0.83 | 0.85 | 0.60 | 0.52 |
| DeepECT + Aug | 0.93 | 0.95 | 0.59 | 0.50 |
| IDEC | 0.86 | 0.85 | 0.58 | 0.53 |
| AE + k-means | 0.70 | 0.77 | 0.52 | 0.48 |
| **CoHiClust** | 0.97 | 0.99 | 0.62 | 0.65 |

We evaluate our method with several datasets of increasing difficulty. We show that CoHiClust outperforms hierarchical clustering models and is among the three best-performing flat clustering methods in terms of similarity with the ground truth classes.In addition, we analyze the constructed hierarchies, which in our opinion are equally important in practical use-cases. Our experiments demonstrate that CoHiClust reveals novel patterns and relations between data points beyond those contained in ground truth classes.

**Comparison with hierarchical clustering methods**    First, we compare CoHiClust with hierarchical baselines. To the best of our knowledge, DeepECT Mautz et al. (2019) is the only hierarchical clustering method based on deep neural networks. Following their experimental setup, we also consider classical hierarchical algorithms evaluated on the latent representation created by the autoencoder and IDEC Guo et al. (2017a). We report the results on two popular image datasets[1]: MNIST and F-MNIST.

---

[1]We intentionally do not use two remaining datasets: USPS which is analogous to MNIST and Reuters which is not an image database.

We run CoHiClust with 16 leaves (4 tree levels) using ResNet18 as a base neural network. The initial model is trained for 50 epochs on MNIST and 100 epochs on F-MNIST using a minibatch size of 256. Next, we perform pruning with an additional 50 and 100 epochs of fine-tuning on MNIST and F-MNIST, respectively, to end up with 10 clusters, which equals the true number of classes for considered datasets. To evaluate the cluster hierarchies against ground truth flat partition, we use dendrogram purity (DP) Kobren et al. (2017); Yang et al. (2019),which attains its maximum value of 1 if and only if all data points from the same class are assigned to some pure sub-tree.

The results summarized in Table 1 demonstrate that CoHiClust outperforms all baselines on both MNIST and F-MNIST datasets. Interestingly, DeepECT benefits from data augmentation in the case of MNIST, while on F-MNIST it deteriorates its performance. All methods except CoHiClust and DeepECT failed completely to create a hierarchy recovering true classes, which confirms that there is a lack of powerful hierarchical clustering methods based on neural networks.

To further evaluate the performance on MNIST and F-MNIST, we measure the normalized mutual information (NMI) and the clustering accuracy (ACC) between the ground truth partition and the flat clustering generated from the tree hierarchy. We exclude the combinations of hierarchical and embedding methods from the comparison due to their poor performance. Instead, we consider two flat clustering methods: IDEC and k-means evaluated on the latent representation generated by the autoencoder. It can be seen from Table 2 that CoHiClust outperforms baseline methods across all datasets and metrics. The disproportion between the results obtained on MNIST and F-MNIST demonstrates that recovering true classes of clothes is a significantly more challenging task than recognizing the hand-written digits.

**Analyzing the clustering hierarchies** Although comparing constructed clustering with the ground truth partition is a widely-used evaluation measure in the literature, it is even more important from a practical point of view to visualize the results and perform their qualitative assessment. Figure 1 presents the clustering tree constructed for F-MNIST (see appendix for the hierarchy of MNIST). Images in nodes show the means calculated over data points assigned to the respective nodes. For every leaf, we also present sample images classified to this cluster and the percentage of images from dominant classes in the group (with a fraction higher than 15%). Visual inspection shows that neighboring leaves contain images with similar patterns, see description in Figure 1 for detailed findings.

The hierarchy also allows us to define the distance $d(a, b)$ between two examples $a, b$ using the number of edges that connect them. We use the average of this score to calculate the similarity between ground truth classes $A$ and $B$ given by $d(A, B) = \frac{1}{Z} \sum_{a \in A} \sum_{b \in B} d(a, b)$, where $Z$ is the number of all pairs. The above distance is small if examples from classes $A$ and $B$ are located in the nearby leaf nodes (on average).

The obtained distance matrix presented in Figure 4 (left) confirms that clusters discovered by Co-HiClust reveal many different patterns than the information encoded in the ground truth classes. While clothes are well separated from other items (number of edges greater than 6), we cannot distinguish particular clothes items from each other (distance between 3 and 4). As can be seen in Figure 1, CoHiClust generated the hierarchy based on the sleeve length or the presence of designs. From this point of view, we can say that the results obtained by CoHiClust are complementary to the ground truth classification. However, we can also observe some similarities with ground truth classification: it is evident that the elements of the following 5 classes have not been spread between clusters: "trousers", "sandals", "sneakers", "bags" and "ankle boots" (we have $d(A, A) < 0$ for these classes).

**Evaluation on large image datasets** In addition to comparing CoHiClust with hierarchical baselines on gray images, we perform the evaluation on large datasets of color images: CIFAR-10, STL-10, ImageNet-Dogs, and ImageNet-10. Since none of the previous hierarchical methods have been examined on these datasets, we use typical agglomerative hierarchical algorithms applied either to raw image data or to the representation generated by our model. Additionally, to compare with a more diverse set of methods, we use the benchmark including flat clustering methods reported in Li et al. (2021a). It is expected that flat clustering methods will perform better because they directly focus on partitioning data into a given number of groups. Hierarchical models, such as CoHiClust, build a hierarchy with the number of leaves exceeding the number of classes, which

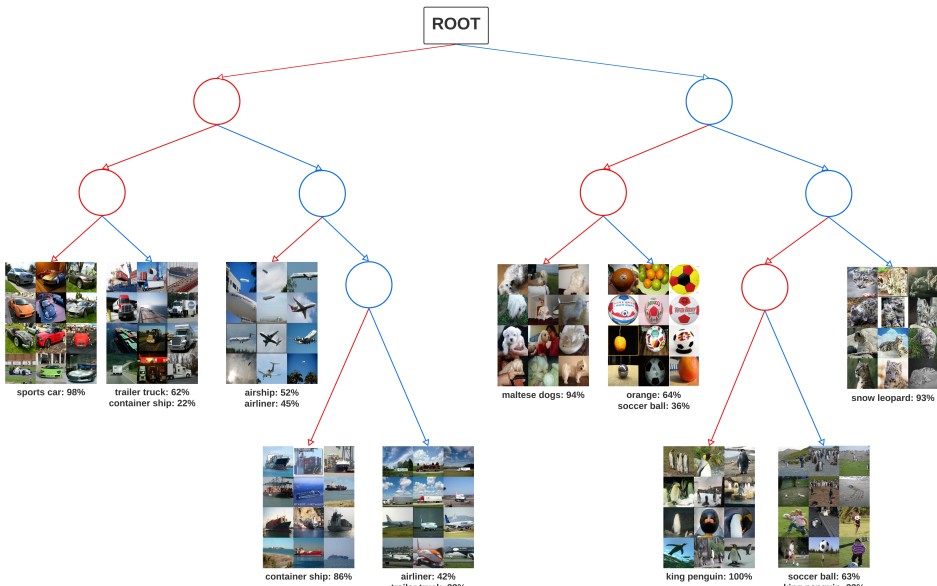

Figure 3: **A tree hierarchy generated by CoHiClust for ImageNet-10.** The left sub-tree contains examples related to the means of transport with neighboring leaves occupied by visually similar classes such as sports cars and trailer trucks (first two leaves from the left) as well as airliners, airships, and container ships (next three leaves). Interestingly, airliners and airships photographed in the sky are assigned to a different cluster than analogical objects located on the ground. In the right sub-tree, CoHiClust almost perfectly clustered examples of two classes: Maltese dog and snow leopard. Surprisingly, images with a ball in the foreground (without people) are grouped together with oranges because of their circular shape. Soccer balls accompanied by the player are considered more similar to the images with penguins (their shapes are also similar). Nevertheless, all images of penguins are assigned to the neighboring leaves.

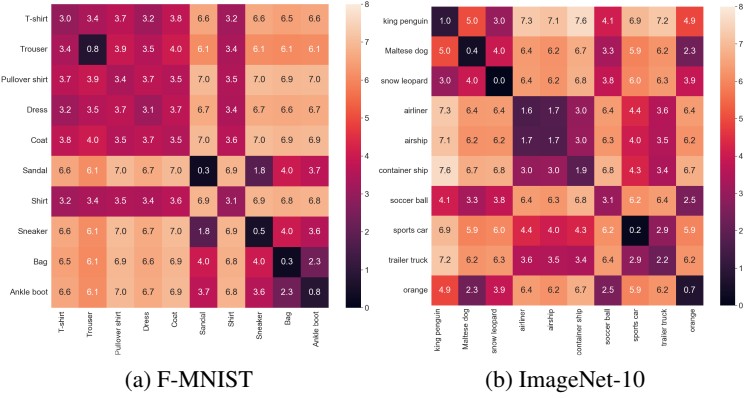

(a) F-MNIST                    (b) ImageNet-10

Figure 4: Distance matrices retrieved from the constructed hierarchies for ground truth classes, see text in the paper for the interpretation.

makes it difficult to obtain so high a resemblance with the ground truth as flat models. Moreover, the optimization of hierarchy is a more difficult and challenging process because final assignments rely on aggregating several binary decisions made by the internal nodes.

Our model uses ResNet34 as a base network, which better suits the complexity of the current images and coincides with the architecture used in Li et al. (2021a). We consider a tree with 16 leaves (4 levels), which is next pruned to obtain the number of clusters equal to the number of ground truth

classes. We train the initial complete tree with the batch size of 256 for 1000 epochs and use 100 epochs after pruning every leaf. To measure the similarity of the constructed partition with the ground truth, we apply three widely-used clustering metrics: normalized mutual information (NMI), clustering accuracy (ACC), and adjusted rand index (ARI). In the appendix, we also show DP of the hierarchies generated by CoHiClust.

Table 3: Comparison with flat (top) and hierarchical (bottom) clustering methods on large image benchmarks.

| Dataset | CIFAR-10 | | | STL-10 | | | ImageNet-10 | | | ImageNet-Dogs | | |
|---|---|---|---|---|---|---|---|---|---|---|---|---|
| Metrics | NMI | ACC | ARI | NMI | ACC | ARI | NMI | ACC | ARI | NMI | ACC | ARI |
| K-means Mac | 0.087 | 0.229 | 0.049 | 0.125 | 0.192 | 0.061 | 0.119 | 0.241 | 0.057 | 0.055 | 0.105 | 0.020 |
| SC Zelnik-Manor & Perona | 0.103 | 0.247 | 0.085 | 0.098 | 0.159 | 0.048 | 0.151 | 0.274 | 0.076 | 0.038 | 0.111 | 0.013 |
| AC Gowda & Krishna (1978) | 0.105 | 0.228 | 0.065 | 0.239 | 0.332 | 0.140 | 0.138 | 0.242 | 0.067 | 0.037 | 0.139 | 0.021 |
| NMF Cai | 0.081 | 0.190 | 0.034 | 0.096 | 0.180 | 0.046 | 0.132 | 0.230 | 0.065 | 0.044 | 0.118 | 0.016 |
| AE Bengio et al. | 0.239 | 0.314 | 0.169 | 0.250 | 0.303 | 0.161 | 0.210 | 0.317 | 0.152 | 0.104 | 0.185 | 0.073 |
| DAE Vincent et al. (2010) | 0.251 | 0.297 | 0.163 | 0.224 | 0.302 | 0.152 | 0.206 | 0.304 | 0.138 | 0.104 | 0.190 | 0.078 |
| DCGAN Radford et al. (2015) | 0.265 | 0.315 | 0.176 | 0.210 | 0.298 | 0.139 | 0.225 | 0.346 | 0.157 | 0.121 | 0.174 | 0.078 |
| DeCNN Zeiler et al. (2010) | 0.240 | 0.282 | 0.174 | 0.227 | 0.299 | 0.162 | 0.186 | 0.313 | 0.142 | 0.098 | 0.175 | 0.073 |
| VAE Kingma & Welling (2013) | 0.245 | 0.291 | 0.167 | 0.200 | 0.282 | 0.146 | 0.193 | 0.334 | 0.168 | 0.107 | 0.179 | 0.079 |
| JULE Yang et al. (2016) | 0.192 | 0.272 | 0.138 | 0.182 | 0.277 | 0.164 | 0.175 | 0.300 | 0.138 | 0.054 | 0.138 | 0.028 |
| DEC Xie et al. (2016) | 0.257 | 0.301 | 0.161 | 0.276 | 0.359 | 0.186 | 0.282 | 0.381 | 0.203 | 0.122 | 0.195 | 0.079 |
| DAC Chang et al. (2017) | 0.396 | 0.522 | 0.306 | 0.366 | 0.470 | 0.257 | 0.394 | 0.527 | 0.302 | 0.219 | 0.275 | 0.111 |
| DDC Chang et al. (2019) | 0.424 | 0.524 | 0.329 | 0.371 | 0.489 | 0.267 | 0.433 | 0.577 | 0.345 | 0.239 | 0.306 | 0.128 |
| DCCM Wu et al. (2019) | 0.496 | 0.623 | 0.408 | 0.376 | 0.482 | 0.262 | 0.608 | 0.710 | 0.555 | 0.321 | 0.383 | 0.182 |
| PICA Huang et al. (2020) | 0.591 | 0.696 | 0.512 | 0.611 | 0.713 | 0.531 | 0.802 | 0.870 | 0.761 | 0.352 | 0.352 | 0.201 |
| CC Li et al. (2021a) | 0.705 | 0.790 | 0.637 | 0.764 | 0.850 | 0.726 | 0.859 | 0.893 | 0.822 | 0.445 | 0.429 | 0.274 |
| Agglom. on raw image | 0.085 | 0.043 | 0.207 | 0.079 | 0.042 | 0.207 | 0.181 | 0.072 | 0.293 | 0.101 | 0.017 | 0.151 |
| Agglom. on pre-trained rep. | 0.350 | 0.503 | 0.255 | 0.464 | 0.477 | 0.321 | 0.530 | 0.532 | 0.343 | 0.252 | 0.092 | 0.276 |
| **CoHiClust** | 0.532 | 0.638 | 0.453 | 0.503 | 0.529 | 0.386 | 0.742 | 0.744 | 0.622 | 0.407 | 0.350 | 0.193 |

The results presented in Table 3 show that CoHiClust outperforms hierarchical baselines with a large margin (bottom rows). Moreover, it is usually among the three best-performing flat methods (top rows). It gives very good results, compared to the baselines, on the subset of the ImageNet dataset achieving 0.742 of NMI. Nevertheless, one should keep in mind that CoHiClust is the only hierarchical method in this comparison, which makes it more challenging to construct flat partition, which resembles the ground truth classes.

To better analyze the results returned by CoHiClust, we plot the constructed clustering hierarchies and distance matrices calculated for ground truth classes. Figure 3 presents and discusses the tree hierarchy constructed for ImageNet-10 (remaining hierarchies together with distance matrices are presented in the appendix). Supporting the analysis with the use of a distance matrix, we can observe that examples of 5 out of 10 classes have not been spread across different clusters (we have $d(A, A) \leq 1$). Airliner and airship classes are the most semantically similar. The class king penguin is the most distinct from the classes representing the means of transport with an average distance greater than 7. Overall, we can see a very high similarity between classes related to the means of transport. The distance between other classes is not as small, which suggests that there are no evident patterns that connect e.g. king penguin, Maltese dog, or orange.

## 5 CONCLUSION AND FUTURE DIRECTIONS

We proposed a contrastive hierarchical clustering model CoHiClust, which suits well to clustering of large-scale image databases. The hierarchical structure constructed by CoHiClust delivers significantly more information about data than typical flat clustering models. In particular, we can inspect the similarity between selected groups by measuring their distance in the hierarchy tree and, in consequence, find super-clusters. Experimental analysis performed on typical clustering benchmarks confirms that the produced partitions have high similarity with ground truth classes. At the same time CoHiClust allows for discovering important patterns which have not been encoded in the class labels.

In future, we plan to apply the constructed model to other unsupervised problems such as anomaly detection. Moreover, we will work on constructing clustering hierarchies which are not restricted to a predefined topology. In particular, inspired by Tanno et al. (2019); Struski et al. (2021), we will focus on models, which automatically adapt the hierarchy structure to a given dataset. Finally, it is an open issue whether it is possible to extend flat clustering models Li et al. (2021a); Hua to the hierarchical ones, which could further improve the clustering metrics.

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

## A  Supplementary qualitative results

In Figures 5, 7, 6 and 8 we illustrate the tree hierarchies constructed by CoHiClust for the remaining datasets. Distance matrices for ground true classes generated from these hierarchies are shown in Figure 9.

**MNIST**   Observe that neighboring leaves contain images of visually similar classes, e.g. 8 and 0; 4 and 9; 1 and 7. Such a property holds also for nodes in the upper levels of the tree – the left sub-tree contain digits with circular shapes, while the digits located in the right sub-tree consist of lines.

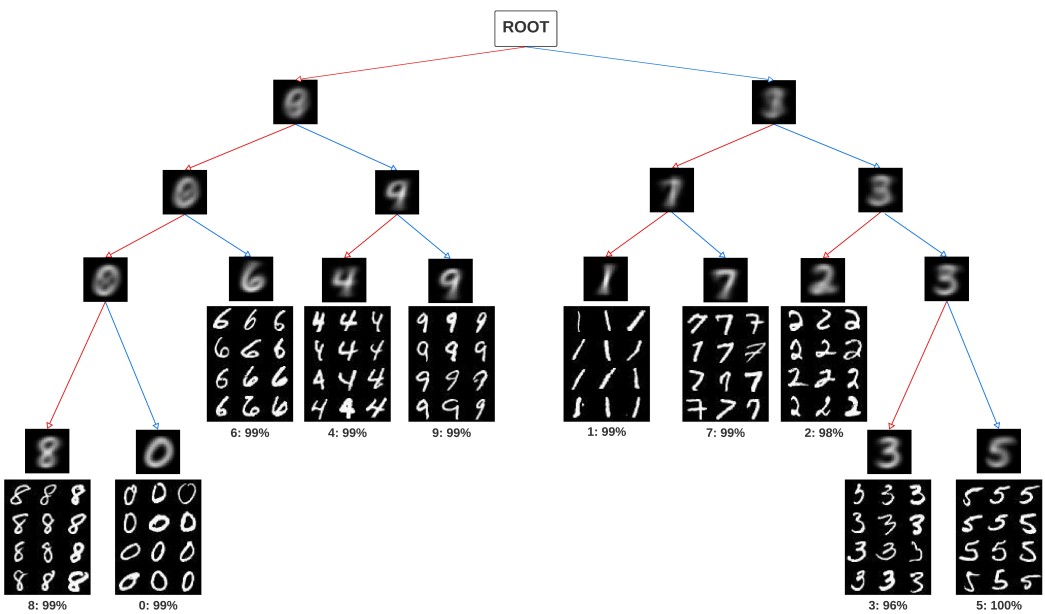

Figure 5: Tree hierarchy constructed for MNIST.

**STL-10**   It is evident that images that present means of transport (car, truck, airplane, ship) were grouped in different subtrees than images with animals (dog, cat, monkey, horse, deer). Only images of birds were mixed up between these subtrees. Birds in the sky or on water background were grouped together with airplanes, while others were assigned to the animal subtree. Such assignments are consistent with our intuition because the shape of the bird and the airplane is similar and the background is the same. The relation of the bird class with other classes is revealed in the distance matrix, see Figure 9. The second row shows a high similarity to airplanes, while the level of similarity to animal classes is almost equal. Ships were considered to be more similar to airplanes than vehicles (cars, trucks), which also confirms the intuitive behavior of CoHiClust. In the case of animal subtree, the model grouped horses and deers in the neighbor leaves, which is also reflected in the distance matrix. Most of the images that presented dogs and cats were also classified in the same subtree containing three leaves, suggesting their similarity.

**CIFAR-10**   The analysis of the hierarchy of CIFAR-10 reveals similar relations as in the case of STL-10 – we have a division into machines and animals in the first level. Moreover, airplanes and birds were considered to be similar classes. In contrast to STL-10, we can observe that CoHiClust created two neighbor groups for animals with white and black fur, respectively. This grouping was associated more with color than with the type of animal. We consider this behavior to be natural, even if it breaks the boundary between ground-truth classes. We emphasize that decisions made by CoHiClust are not based on classes but only on the type of augmentation. Nevertheless, CoHiClust was able to create individual clusters dominated by the following classes: frogs, deers, horses, trucks, and cars. By analyzing the distance matrix, we can observe pairs of similar classes: plane and ship, dog and horse, etc.

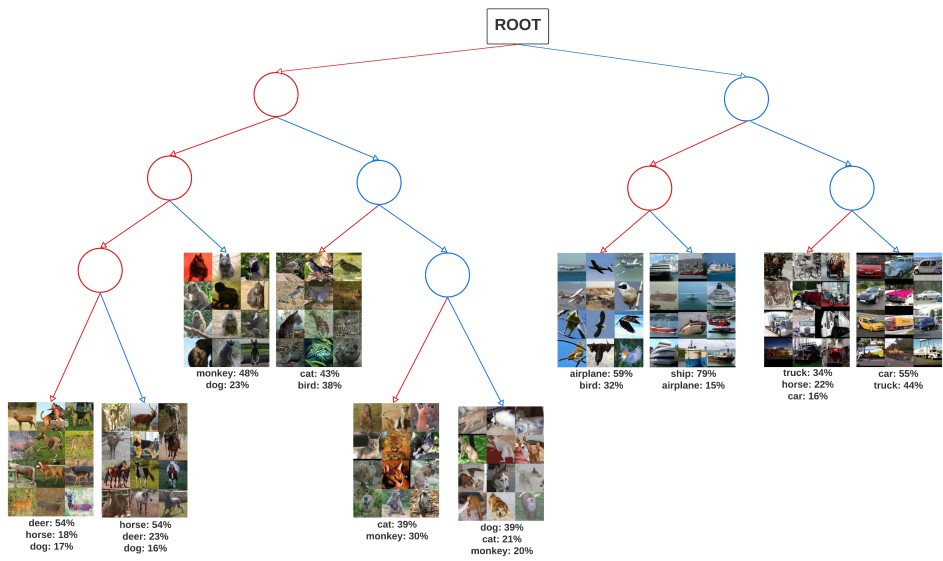

Figure 6: Tree hierarchy constructed for STL-10.

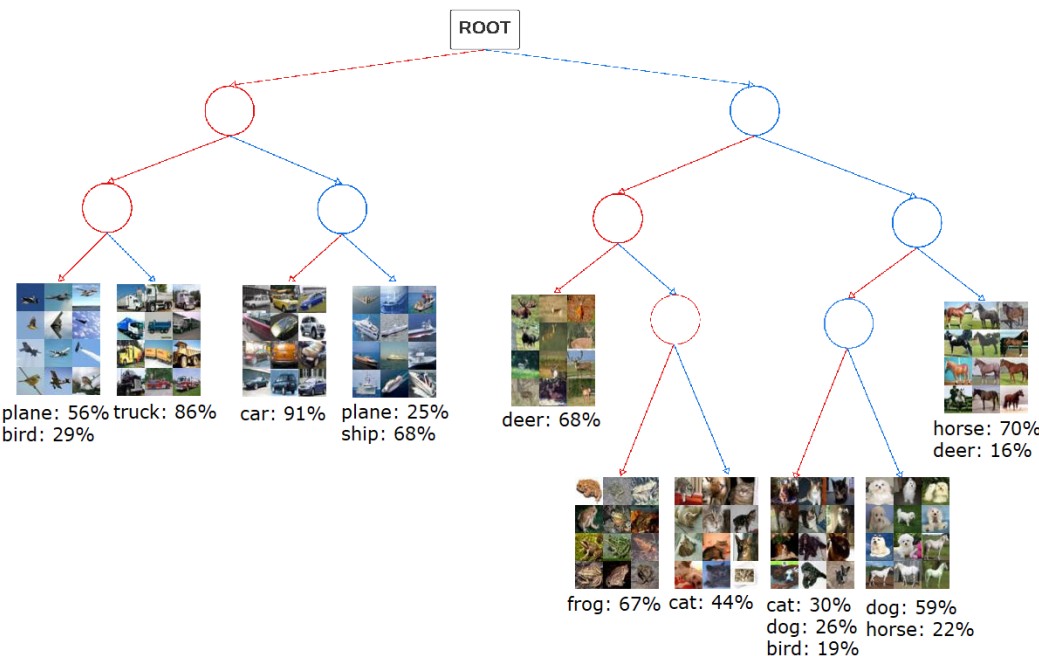

Figure 7: Tree hierarchy constructed for CIFAR-10.

# B  ANALYSIS OF THE COHICLUST MODEL

In this section, we analyze selected properties of CoHiClust including the choice of backbone network, the form of the loss function, techniques for reducing the number of leaves and possible training strategies. If not stated otherwise, the experiments were run on CIFAR-10 dataset, which we consider the most representative.

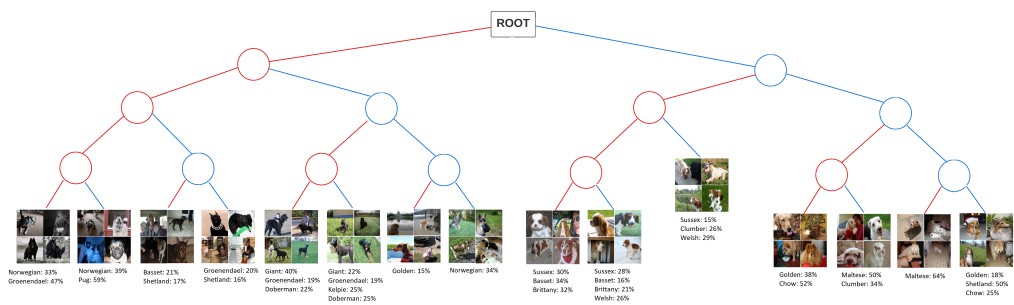

Figure 8: Tree hierarchy constructed for ImageNet-Dogs.

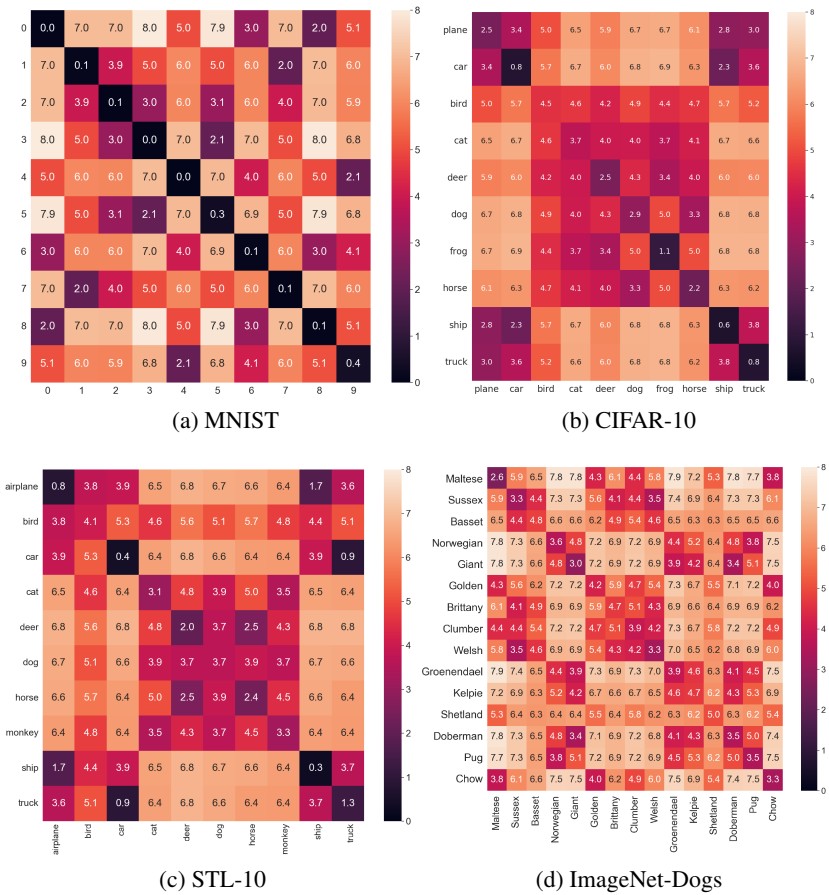

Figure 9: Distance matrices retrieved from the constructed hierarchies for ground truth classes.

### B.1 RELIANCE ON BACKBONE NETWORK

In Table 4, we show how the selection of the architecture for the base network $g$ influences the performance of CoHiClust. As can be seen, the difference in the results is marginal.

### B.2 ANALYSIS OF LOSS FUNCTION

Next, we explain the influence of particular components of CoHiClust loss function. Namely, we remove one or both regularization terms from the loss function and report the results. As shown in Table 5, $R_2$ regularization (NT-XENT) has a notable impact on the metrics, since it improves the

Table 4: Comparison of ResNet architectures.

| Dataset | CIFAR-10 | | | STL-10 | | |
|---|---|---|---|---|---|---|
| Metrics | NMI | ACC | ARI | NMI | ACC | ARI |
| ResNet18 | 0.481 | 0.510 | 0.362 | 0.503 | 0.529 | 0.386 |
| ResNet34 | 0.532 | 0.638 | 0.453 | 0.498 | 0.467 | 0.366 |

representation of data. However, the obtained model has a tendency to build smaller tree and reduce clusters during training (third row). The influence of $R_1$ (entropy) on the quality of clustering is marginal, but this term is crucial for balancing decisions made by the internal nodes and preventing from clusters reduction (second row). Combining both regularization terms allows us to build a complete hierarchy with satisfactory metrics.

Table 5: Ablation study of CoHiClust loss function performed on CIFAR-10.

| | DP | NMI | ACC | ARI | # Clusters |
|---|---|---|---|---|---|
| CoHiLoss | 0.187 | 0.271 | 0.289 | 0.172 | 4 |
| CoHiLoss + R1 | 0.192 | 0.260 | 0.296 | 0.181 | 12 |
| CoHiLoss + R2 | 0.444 | 0.519 | 0.613 | 0.425 | 9 |
| CoHiLoss + R1 + R2 | 0.446 | 0.532 | 0.639 | 0.453 | 16 |

Following the above analysis, we observe that CoHiClust can automatically detect the number of clusters if we eliminate the entropy regularization. More precisely, $R_2$ is essential for delivering appropriate representation of data, while $R_1$ is only used to maintain the requested number clusters. If we are interested in generating hierarchy of the size automatically adjusted to a given dataset, then we have to put $\beta_1 = 0$ and $\beta_2 = 1$.

### B.3 ALTERNATIVE DEFINITIONS OF THE LOSS FUNCTION

Let us recall that the proposed CoHiLoss contains the square root to enable the model to explore possible tree paths, see equation 1. If we do not use the square root, we encourage the model to binarize decisions in nodes, which can lead to stacking in local minima at the initial phase of training. The results in Table 6 show that the model with square root obtains better metrics than the one without the square root. It partially confirms our initial hypothesis that using the square root influences positively on the exploration of the tree.

Table 6: The influence of the square root in CoHiLoss.

| | DP | NMI | ACC | ARI |
|---|---|---|---|---|
| CoHiClust w/ square root | 0.402 | 0.446 | 0.425 | 0.297 |
| CoHiClust + w/o square root | 0.342 | 0.410 | 0.425 | 0.283 |
| CoHiClust w/ square root (pruned) | 0.447 | 0.532 | 0.638 | 0.453 |
| CoHiClust w/o square root (pruned) | 0.387 | 0.457 | 0.572 | 0.373 |

### B.4 COMPARING TECHNIQUES FOR REDUCING THE NUMBER OF LEAVES

Complete hierarchy is a useful tool to analyze the relationships between data. To generate the hierarchy with smaller number of leaves we have the following options:

- Eliminate entropy regularization from the loss function ($\beta_1 = 0$). In this case, the model automatically detects the number of groups.
- Reduce the least significant leaves simultaneously with fine-tuning the model. Fine-tuning is performed during pruning and after obtaining the final model (our default strategy).
- Delete the least significant leaves without fine-tuning. This corresponds to one of typical pruning strategies used in decision trees.
- Delete the least significant and next train the model from scratch but with the new tree structure.

Table 7 shows the clustering metrics for the obtained models evaluated on CIFAR-10. As can be seen, pruning combined with fine-tuning gives the best results (third row), improving significantly the clustering structure delivered by the complete hierarchy (second row). Slightly lower metrics are obtained when we retrain the reduced tree from scratch (fifth). The model without fine-tuning (fourth row) also gives higher scores than the base model without pruning, however the gain in clustering metrics is not so high. This shows that fine-tuning the pruned model is an essential step leading in adjusting the hierarchy to a given number of groups. It is notable that the model without entropy regularization ($\beta_1 = 0$) is able to reasonably well detect the number of clusters as well as to produce satisfactory results. It obtains better clustering metrics than the model with pruning but without fine-tuning.

Table 7: Comparing techniques for reducing the number of leaves.

| | DP | NMI | ACC | ARI | #Clusters |
|---|---|---|---|---|---|
| CoHiClust ($\beta_1 = 0$) | 0.403 | 0.486 | 0.569 | 0.390 | 9 |
| CoHiClust (w/o pruning) | 0.402 | 0.446 | 0.425 | 0.297 | 16 |
| CoHiClust (pruning and fine-tuning) | 0.447 | 0.532 | 0.638 | 0.453 | 10 |
| CoHiClust (pruning only) | 0.404 | 0.461 | 0.570 | 0.373 | 10 |
| CoHiClust (pruning and retraining from scratch) | 0.436 | 0.519 | 0.616 | 0.437 | 10 |

### B.5 END-TO-END TRAINING VS. PRE-TRAINED REPRESENTATIONS

Finally, we examine whether end-to-end training is crucial for the performance of CoHiClust. For this purpose, we first pretrain the base network $g$ using NT-Xent loss, which corresponds to the typical SimCLR model Chen et al. (2020). Next, we add the projection head $\pi$ to $g$ and train CoHiClust using $\text{CoHiLoss} + R_2$ loss. As can be seen in Table 8 the model trained in an end-to-end manner obtains significantly better scores.

Table 8: Training CoHiClust end-to-end vs. training CoHiClust on representation taken from pre-trained SimCLR.

| | NMI | ACC | ARI |
|---|---|---|---|
| end-to-end training | 0.532 | 0.638 | 0.453 |
| pretrained representation | 0.434 | 0.480 | 0.297 |

## C DENDROGRAM PURITY FOR DATASETS OF COLOR IMAGES

In Table 9, we present dendrogram purity obtained by CoHiClust for datasets of color images. This benchmark can be used to compare other hierarchical clustering models.

Table 9: Dendrogram purity on color images.

| | DP |
|---|---|
| CIFAR-10 | 0.447 |
| STL-10 | 0.345 |
| ImageNet-10 | 0.613 |
| ImageNet-Dogs | 0.198 |

