# OpenReview forum: "Contrastive Hierarchical Clustering"
_ICLR.cc/2023/Conference — Submitted to ICLR 2023_

### Official Review · Reviewer_8rQe · 2022-10-23

**Confidence:** 5
**Clarity, Quality, Novelty And Reproducibility:** The paper is written clearly and easy…
**Correctness:** 4
**Technical Novelty And Significance:** 2
**Empirical Novelty And Significance:** 2
**Recommendation:** 3

**Strength And Weaknesses:**

Strength:
1. By incorporating hierarchical clustering into the deep model, the proposed method could be easily applied to large-scale data.
2. The presentation is easy to follow.

Weaknesses:
1. My major concern is that the novelty is insufficient. It seems that the proposed method only uses some self-supervised methods for representation learning and combines them with some hierarchical clustering techniques. Although there are no specific models for performing hierarchical clustering under the contrastive learning framework, the existing techniques are somehow ready to apply like Contrastive Clustering. The novelty in terms of the solution technique is thus limited.
2. The authors claimed that the proposed method could achieve comparable performance to the SOTA clustering methods, but the results from Table 3 show a large gap between the proposed method and the SOTA baselines.
3. From Table 1 and Table 2, I notice that the DeepECT + Aug could achieve similar performance to the proposed method. So here comes the question: In the proposed method, is the augmentation technique from self-supervised learning the primary reason for boosting the performance? I wonder about the influence of different components of the proposed method on the final results. However, I didn't see any ablation studies about this.

**Summary Of The Paper:**

This paper focuses on hierarchical clustering which could build a hierarchy of clusters. The proposed method leverage some self-supervised learning techniques to obtain a high-level representation that is used for tree hierarchy building. As the proposed method is a deep model trained in an end-to-end manner, it could be applied to the large-scale dataset for hierarchical clustering. Some qualitative examples show that the proposed method can yield a good tree hierarchy.

**Summary Of The Review:**

This paper proposes a new deep hierarchical clustering model for large-scale data. The paper is well-written. However, the contribution and novelty of this paper are limited.

---

> ### Author Response · Authors · 2022-11-11
> **Authors' response to Reviewer 8rQe (part 2)**
>
> **Q3: From Table 1 and Table 2, I notice that the DeepECT + Aug could achieve similar performance to the proposed method. I wonder about the influence of different components of the proposed method on the final results.**
>
> We agree that DeepECT is an important technique, which was one of our inspirations, but it does not perform similarly to CoHiClust. In the case of MNIST the difference in ACC is 0.04, but CoHiClust obtains 0.99 so it is impossible to further increase its performance. A similar situation appears for other metrics, where CoHiClust obtains 0.97. In the case of FMNIST, CoHiClust gives 0.65 of ACC, while DeepECT + Aug gives only 0.5.
>
> To explain the influence of particular components of CoHiClust, we performed the ablation study on CIFAR-10. As shown in the table below (or see Table  5 in the revised version), R2 regularization (NT-XENT) has a notable impact on the metrics.
>
> |                    | NMI   | #Clusters |
> |--------------------|-------|-----------|
> | CoHiLoss           | 0.271 | 4         |
> | CoHiLoss + R1      | 0.260 | 12        |
> | CoHiLoss + R2      | 0.519 | 9         |
> | CoHiLoss + R1 + R2 | 0.532 | 10        |
>
> However, this model has a tendency to build smaller trees and reduce clusters during training. If we are interested in building a complete hierarchy with a given number of clusters, we should use R1 regularization to balance decisions made by internal nodes. On the other hand, eliminating R2 regularization from the loss function drastically lower the metrics because the tree is built on inappropriate representation (first two rows). Combining both regularization terms allows us to build a complete hierarchy with satisfactory metrics. We add this analysis to the revised version in the appendix.
>
> We also explored possible modifications of our loss function and eliminated the square root from the CoHiLoss, eq. 1. The results in the table below (or see Table 6 in the revised version) show that the application of the square root improved the results. Our hypothesis is that using the square root influences positively the exploration of the tree in the initial stage.
>
> |                                    | NMI   |
> |------------------------------------|-------|
> | CoHiClust w/ square root           | 0.446 |
> | CoHiClust + w/o square root        | 0.410 |
> | CoHiClust w/ square root (pruned)  | 0.532 |
> | CoHiClust w/o square root (pruned) | 0.457 |
>
> All ablation studies are included in appendix B of the revised version.

---

> ### Author Response · Authors · 2022-11-13
> **Authors' response to Reviewer 8rQe (part 1)**
>
> Thank you for the valuable and constructive criticism.
>
> **Q1: Novelty: It seems that the proposed method only uses some self-supervised methods for representation learning and combines them with some hierarchical clustering techniques. Although there are no specific models for performing hierarchical clustering under the contrastive learning framework, the existing techniques are somehow ready to apply like Contrastive Clustering. The novelty in terms of the solution technique is thus limited.**
>
> We argue that the construction of CoHiClust is not straightforward. Indeed, we follow a self-supervised learning principle, which indicates how to construct pairs of similar examples, but we manage these pairs in our own way. We are not aware of any work, which used a self-supervised learning approach in hierarchical clustering.
>
> Existing contrastive (flat) clustering techniques do not require modeling the hierarchy but only assignments to clusters, which can be considered a simpler task. In particular, since CoHiClust uses probabilistic decisions in nodes, we needed to design contrastive loss based on probabilistic inputs. To the best of our knowledge, recent contrastive losses use vector inputs, which are more convenient because the dimensionality of such space can be arbitrary. In our case, we are restricted by the number of nodes.
>
> We agree that there is room for designing new hierarchical clustering models and we hope that our paper will be an inspiration for other authors to construct such models. However, we cannot agree that the construction of deep hierarchical models is a simple adaptation of existing techniques.
>
> **Q2: The authors claimed that the proposed method could achieve comparable performance to the SOTA clustering methods, but the results from Table 3 show a large gap between the proposed method and the SOTA baselines.**
>
> Data clustering is an unsupervised technique that makes it difficult or even impossible to evaluate its performance. In our opinion equally important to reporting the clustering metrics like NMI is analyzing the results. NMI gives only the level of similarity with the ground truth, which may not be exactly what we are looking for in clustering. Our paper contains the analysis of the obtained results, which confirms that CoHiClust finds important patterns in data, see the caption of Figures 1 and 3 and our analysis in the paper.
>
> We decided to report all metrics to be consistent with the literature. According to the metrics, CoHiClust is only worse than PICA and CC, which we consider as a very good result because all baseline techniques represent flat models. To address the Reviewer's comment, we weaken the statement that our method gives comparable results to the SOTA clustering methods in the revised version.
>
> To extend our benchmark on color images and show that CoHiClust works significantly better than existing hierarchical baselines, we perform additional experiments with the use of typical agglomerative clustering. In the first case, agglomerative clustering was applied to raw image data, while in the second case, it was evaluated on the representation taken from our contrastive model. The results in the table below (or see Table 3 in the revised version) show a big gap between our approach and classical techniques even if both algorithms were run on the same representation.
>
> |                             | CIFAR-10 | STL-10 | IN-10 | IN-Dogs |
> |-----------------------------|----------|--------|-------|---------|
> | Agglom. on raw image        | 0.085    | 0.079  | 0.181 | 0.101   |
> | Agglom. on pre-trained rep. | 0.350    | 0.464  | 0.530 | 0.252   |
> | CoHiClust                   | 0.532    | 0.503  | 0.742 | 0.407   |

---

### Official Review · Reviewer_4K5k · 2022-10-23

**Confidence:** 4
**Clarity, Quality, Novelty And Reproducibility:** Please refer to Summary Of The Review.
**Correctness:** 3
**Technical Novelty And Significance:** 2
**Empirical Novelty And Significance:** 3
**Recommendation:** 5

**Strength And Weaknesses:**

Please refer to Summary Of The Review.

**Summary Of The Paper:**

Please refer to Summary Of The Review.

**Summary Of The Review:**

This paper applies the deep neural networks technology to clustering and establishes a novel clustering model which is dubbed CoHiClust. In this model, a contrastive learning method is employed to create the base network (i.e. a binary tree).so that the original data are represented on a higher-level. Then two regularization strategies are used to make the base neural network train with the arbitrary number of leaves. The cluster assignment of each sample is achieved by aggregating binary decisions. Besides, to create a tree with a fixed number of leaves, a pruning step is added. The experiments conducted on large-scale image datasets shows the CoHiClust better clustering performance compared to some existing deep and flat clustering algorithms.

Deep clustering is a research hotspot in machine learning in recent years, so the topic in this paper is up-to-date and meaningful. The work of this paper has a certain degree of innovation, but there are some issues or suggestions that I want to point out.

1) Section 1 lacks a sound analysis of research motivation. Why combine deep neural network with hierarchical clustering? A relevant algorithm DeepECT has been proposed, so why further research on this topic? Why is it important to examine the algorithm to large-scale image datasets?
2) The specific meaning of parameters \beta_1and \beta_2 in Eq. (1) and the basis for their values should be added. Further, in the experimental part there should be an analysis of the impact of their different values on CoHiClust performance.
3) In terms of experimental design, why only large-scale image datasets are used to test the performance of used algorithms? Can the algorithm proposed in this paper be applied to other types of datasets? The key point is that the proposed CoHiClust model does not seem to have a special design for large-scale image data.
4) What does the abbreviation AE in Table 1 refer to? Classical hierarchical clustering algorithm?
5) When analyzing the clustering hierarchies on page 7, why not compare CoHiClust with the classical hierarchical clustering algorithms or the latest ones? To further evaluate the validity of this paper work, I strongly recommend adding this comparison.
6) The readability of the tables must be improved. For example, the best values can be in bold.
7) I highly recommend to the authors to add open issues and future directions of their work.

To sum up, this paper need more revisions for its presentation and more experimental work.

---

> ### Author Response · Authors · 2022-11-11
> **Authors' response to Reviewer 4K5k (part 2)**
>
>
> **Q3: Why only large-scale image datasets are used? Can the algorithm proposed in this paper be applied to other types of datasets?**
>
> We examined the performance of CoHiClust on diverse datasets including simple MNIST datasets as well as subsets of large ImageNet database. We did not restrict only to large-scale image data but emphasized that our method can also be successfully used for such data, which in our opinion is a more challenging problem. DeepECT, our main hierarchical baseline, has been tested only on small datasets, such as MNIST and FMNIST.
>
> Our approach follows the typical self-supervised learning principle and can be applied to the same datatypes as other self-supervised techniques. In consequence, it can be applied to text data as well as to graphs of chemical molecules.
>
> **Q4: What does the abbreviation AE in Table 1 refer to?**
>
> AE in Table 1 refers to the pre-trained autoencoder from DeepECT paper. In this case, classical hierarchical algorithms (bisecting k-means or agglomerative hierarchical clustering with a specific linkage function) were applied to the representation taken from AE trained on a given dataset.
>
> **Q5: Why not compare CoHiClust with the classical hierarchical clustering algorithms or the latest ones in Table 3?**
>
> To address the Reviewer's comment, we perform additional experiments with the use of typical agglomerative hierarchical clustering on datasets from Table 3. In the first case, agglomerative clustering was applied to raw image data, while in the second case, it was evaluated on the representation taken from our contrastive model. The results in  the table below (or see Table 3 in the revised version) show a big gap between our approach and classical techniques even if both algorithms were run on the same representation.
>
> |                             | CIFAR-10 | STL-10 | IN-10 | IN-Dogs |
> |-----------------------------|----------|--------|-------|---------|
> | Agglom. on raw image        | 0.085    | 0.079  | 0.181 | 0.101   |
> | Agglom. on pre-trained rep. | 0.350    | 0.464  | 0.530 | 0.252   |
> | CoHiClust                   | 0.532    | 0.503  | 0.742 | 0.407   |
>
> **Q6: The readability of the tables must be improved. For example, the best values can be in bold.**
>
> As explained in the introduction (first paragraph) and experiments, comparing obtained partitions with ground-truth classes is not a primary goal of clustering. The aim of clustering is to deliver information about data and not to perform classification. In consequence, we focus more on analyzing constructed hierarchies than clustering metrics which are not so important in practical applications. However, to be consistent with the literature concerning data clustering, we decided to keep the metrics in paper. At the same time, boldfacing the highest values is misleading in our opinion. We would like to encourage the reader to focus more on our qualitative results than comparing metrics.
>
> **Q7: I highly recommend to the authors to add open issues and future directions of their work.**
>
> We extended the last section of the paper and indicated possible future directions and open issues.

---

> ### Author Response · Authors · 2022-11-13
> **Authors' response to Reviewer 4K5k (part 1)**
>
> Thank you for your detailed remarks.
>
> **Q1: Section 1 lacks a sound analysis of research motivation. Why combine deep neural
> network with hierarchical clustering? A relevant algorithm DeepECT has been proposed, so
> why further research on this topic? Why is it important to examine the algorithm to large-scale image datasets?**
>
> We would like to thank the referee for pointing out this questions. Hierarchical clustering is an extremely important technique in data analysis and machine learning. The hierarchy of clusters delivers significantly more information about relations between clusters than flat partitions. However, typical hierarchical algorithms were not designed to cluster image data, which results in their poor performance in this domain. To confirm this thesis we additionally applied typical agglomerative hierarchical clustering to datasets from Table 3. In the first case, agglomerative clustering was applied to raw image data, while in the second case, it was evaluated on the representation taken from our contrastive model. The results in the table below (or see Table 3 in a revised version) show a big gap between our approach and classical techniques even if both algorithms were run on the same representation.
>
>
> |                             | CIFAR-10 | STL-10 | IN-10 | IN-Dogs |
> |-----------------------------|----------|--------|-------|---------|
> | Agglom. on raw image        | 0.085    | 0.079  | 0.181 | 0.101   |
> | Agglom. on pre-trained rep. | 0.350    | 0.464  | 0.530 | 0.252   |
> | CoHiClust                   | 0.532    | 0.503  | 0.742 | 0.407   |
>
> DeepECT was the first step in overcoming this problem. The authors showed that it is possible to significantly improve the performance on small gray image data (MNIST and FMNIST), but the algorithm was not designed to cluster larger databases of color images. Making use of a self-supervised learning approach, we are able to further boost the performance on MNIST and FMNIST as well as obtain promising results on datasets of color images. To the best of our knowledge, this is the first hierarchical clustering model using deep neural networks, which has been applied to databases of color images. We included this motivation in the second paragraph of Section 1.
>
> **Q2: The specific meaning and analysis of parameters $\beta_1$ and $\beta_2$ in Eq. (2) should be added.**
>
> We added a better description of $\beta_1$ and $\beta_2$ after equation (2) and included the analysis of regularization parameters in the appendix (Table 5). We also put the results in the table below.
>
> |                    | NMI   | #Clusters |
> |--------------------|-------|-----------|
> | CoHiLoss           | 0.271 | 4         |
> | CoHiLoss + R1      | 0.260 | 12        |
> | CoHiLoss + R2      | 0.519 | 9         |
> | CoHiLoss + R1 + R2 | 0.532 | 10        |
>
> The parameter $\beta_1$ is the weight of the entropy regularization which prevents the model from reducing clusters. It can slightly improve the model performance, but its main role is to maintain the complete hierarchy. If we eliminate this term, we let the model automatically detect the number of clusters, which can also be of practical importance in real use cases. The parameter $\beta_2$  is the weight of NT-XENT regularization, which is responsible for improving the representation of data. This term is crucial for generating a high-quality hierarchy.

---

### Official Review · Reviewer_PDcK · 2022-10-24

**Confidence:** 3
**Correctness:** 3
**Technical Novelty And Significance:** 3
**Empirical Novelty And Significance:** 2
**Recommendation:** 6

**Clarity, Quality, Novelty And Reproducibility:**

The clarity is good, and the reproducibility is guaranteed due to the simplicity of the proposed method and the codes attached in the Appendix. The reviewer feels the novelty of the proposed binary tree + contrastive loss is ok for ICLR.

**Strength And Weaknesses:**

The reviewer believes the studied direction and problem are very important for deep learning and representation learning. This paper proposed an interesting perspective to model the classification as a binary tree decision process. A proper contrastive objective is proposed to make the framework trainable. Overall the paper is easy to follow.

Extensive results are conducted in MNIST, FMNIST, CIFAR-10, STL-10, IN-10, and IN-Dogs and show the strength of the proposed hierarchical method compared to other hierarchical clustering and the classification gap between the "flat" clustering method, which are good for readers to understand what the proposed method has achieved. However, the qualitative results are a bit limited for people to understand if the approach really can generate proper hierarchies. Not many ablation studies are included.

The reviewer is unsure if Mautz et al is the only related hierarchical clustering proposed in the deep learning era, as mentioned in the introduction. Some related methods could be developed in understanding or utilizing class hierarchy papers, such as [1] Large-Scale Few-Shot Learning: Knowledge Transfer With Class Hierarchy (***not need to cite***). The reviewer feels some application papers in CV and NLP may have developed related methods to hierarchical clustering, including agglomerative clustering.

The proposed method assumes fixed height, which is a major limitation. Also, the proposed pruning strategy requires fine-tuning, which makes this step pretty ad-hoc because we can also set up a new structure and re-train from scratch. No ablation studies have been conducted to prove the pruning strategy is effective for improving performance or helping find tree hierarchy.

According to Table 3, there are big gaps between the proposed clustering method and the previous ones, which is ok if the author can prove the proposed clustering alg can help find class hierarchy in large-scale datasets. However, the current experiments are limited to ImageNet-10 and ImageNet-Dogs. Training on ImageNet and comparing standard contrastive learning results can help here.

**Summary Of The Paper:**

This paper proposed a hierarchical clustering model with a designed contrastive objective. It decomposes the classification into a series of the decision process as a binary tree. To make the structure more flexible, a pruning strategy is developed. The results on CIFAR-10, STL-10, IN-10, and IN-Dogs show the proposed method still has some performance gap with the SOTA "flat" clustering methods. But, the results of MNIST and FMNIST show the proposed method beats other hierarchical clustering methods.

**Summary Of The Review:**

In general, the reviewer likes the direction, and the proposed method can bring some fresh air to the community. However, the reviewer hopes the author can address the abovementioned concerns, especially about the new qualitative results on large-scale datasets and the ablation studies in the pruning strategy.

---

> ### Author Response · Authors · 2022-11-11
> **Authors' response to Reviewer PdCK (part 2)**
>
> **Q3: The proposed method assumes fixed height, which is a major limitation. The proposed pruning strategy requires fine-tuning. No ablation studies have been conducted to prove the pruning strategy is effective.**
>
> Our goal was to build a complete hierarchy with the requested tree height. If one would like to have a model which detects the number of clusters automatically, it is sufficient to change the parameters of the regularization terms. As shown in Table 5 in the revised version, the model with both regularization terms returns a complete hierarchy with very good metrics after our pruning (last row). Eliminating R1 regularization (entropy) has a marginal effect on the metrics, but allows the model to detect the number of clusters automatically. R1 was designed specifically to balance decisions made by internal nodes. In consequence, CoHiClust is not limited to the fixed height but can detect the number of clusters automatically if one expects such a behavior.
>
> |                    | NMI   | #Clusters |
> |--------------------|-------|-----------|
> | CoHiLoss           | 0.271 | 4         |
> | CoHiLoss + R1      | 0.260 | 12        |
> | CoHiLoss + R2      | 0.519 | 9         |
> | CoHiLoss + R1 + R2 | 0.532 | 10        |
>
>
> Setting up a new structure and re-training from scratch, as suggested by the Reviewer, is not straightforward because it is not clear what tree structure should be used for training. In our approach, we start from the complete tree as we do not have any background knowledge about data. Next, in the proposed pruning, the final tree structure is generated automatically (we reduce the least significant leaves).
>
> The pruning strategy proposed in the paper allows us to obtain the best values of clustering metrics, but this is not the only possible strategy. One alternative is to eliminate the fine-tuning step. More precisely, we reduce the leaves with the lowest number of examples and reassign them to the parent node. As can be seen in the table below (or see Table 7 in the revised version), the pruning with fine-tuning (default strategy) presents the best performance, but the model without entropy regularization also produces good results. The model without fine-tuning obtains the worst performance. It shows that fine-tuning is an essential part of the pruning phase.
>
>
> |                                     | NMI     | #Clusters |
> |-------------------------------------|-------|-----------|
> | CoHiClust (w/o entropy)             | 0.486  | 9         |
> | CoHiClust (pruning and fine-tuning) | 0.532  | 10        |
> | CoHiClust (pruning only)             | 0.461 | 10        |
>
> **Q4: The current experiments are limited to ImageNet-10 and ImageNet-Dogs. Training on ImageNet and comparing standard contrastive learning results can help here.**
>
> A major advantage of CoHiClust over flat clustering algorithms is that it brings extra information about relations between clusters. This information can be either retrieved from the distance matrices or from visualization of the tree. We discuss the hierarchies for FMNIST (caption below Figure 1) and ImageNet-10 (caption below Figure 3 and the paragraph at the bottom of page 7). We believe that the hierarchical information regarding the relations between particular groups of data is beneficial for practitioners. This is confirmed by the popularity of hierarchical algorithms in bioinformatics and chemistry. Typical flat clustering methods can often obtain better metrics but cannot deliver such rich information about clusters. Clustering is unsupervised (we do not have ground truth) so higher metrics do not have to mean better clustering.
>
> We partially agree that evaluating CoHiClust on ImageNet could be of interest and there are no conceptual or computational obstacles to running CoHiClust on ImageNet. However, DeepECT, a previous hierarchical method, was only evaluated on FMNIST and MNIST, while we applied CoHiClust to significantly more challenging datasets. This is a huge step forward considering previous works.

---

> > ### Comment · Reviewer_PDcK · 2022-11-15
> > **Reviewer Response**
> >
> > Hi Authors,
> >
> > I thank you for your efforts and rebuttal. Based on the new results, I have the following concerns/questions about the manuscript. If you can provide your insights, I'd like to appreciate it.
> >
> > 1. For the new qualitative results in Appendix A, it seems the proposed method can well classify between "animal" and "machines" but performs poorly to classify the fine-grained categories. Do I understand correctly? If so, I feel a simple cluster algorithm can partition the two supercategories for both CIFAR and STL.
> >
> > 2. For Figures 6 and 7, I feel there is not necessary to have a third layer. Are there any benefits have the third layer here? For all the qualitative results, I did not feel the strong advantages of the proposed method for finding "extra information about relations between clusters".
> >
> > 3. Regarding the pruning strategy, I am personally curious how the performance would change if you prune the nodes, fix the hierarchy, and re-train from scratch, i.e. CoHiClust (pruning and re-train)
> >
> > 4. we will definitely consider your computation resource, and the DeepECT is applied to fewer datasets. But, since one of the claims is finding cluster relationships, which is more interested in large-scale datasets than only 10 classes datasets.

---

> > > ### Author Response · Authors · 2022-11-17
> > > **Authors' response to Reviewer PdCK**
> > >
> > > We appreciate your quick response. We updated the version of the manuscript, where we described the hierarchies obtained for MNIST, STL-10 and CIFAR-10. Moreover, we included the clustering results of the model, which was retrained after pruning. We hope that these modifications will provide better insights into the proposed model.
> > >
> > > We answer all remarks raised by the Reviewer below:
> > >
> > > **Q1: For the new qualitative results in Appendix A, it seems the proposed method can well classify between "animal" and "machines" but performs poorly to classify the fine-grained categories. Do I understand correctly? If so, I feel a simple cluster algorithm can partition the two supercategories for both CIFAR and STL.**
> > >
> > > We would like to emphasize that we do not expect a clustering algorithm to perform fine-grained classification. It is not the role of clustering. Clearly, we observe that the decisions made by CoHiClust frequently coincide with ground-truth classes, but in many times, CoHiClust makes decisions based on different patterns. In the revised version, we described the hierarchies found by CoHiClust for MNIST, CIFAR-10 and STL-10. We refer the Reviewer to appendix A, where we give detailed explanations of our findings.
> > >
> > > To show that simple hierarchical algorithms cannot discover the clustering sturcure comparable to CoHiClust, we evaluated algomerative hierarchical clustering on color images. The results shown in Table 3 confirm that simple algorithms perform worse than CoHiClust.
> > >
> > > **Q2: For Figures 6 and 7, I feel there is not necessary to have a third layer. Are there any benefits have the third layer here? For all the qualitative results, I did not feel the strong advantages of the proposed method for finding "extra information about relations between clusters".**
> > >
> > > In hierarchical clustering, we can view the splitting of data at various granularity levels. In typical clustering, we perform splitting until we obtain single-element clusters. In our case, we fix a maximal tree heights. In both cases, we can restrict to a given number of clusters. To compare our results with ground truth, we restricted our attention to 10 clusters for STL-10 and CIFAR-10. The major advantage of hierarchical algorithms is that we are not restricted to a given number of clusters or levels.
> > >
> > > **Q3: Regarding the pruning strategy, I am personally curious how the performance would change if you prune the nodes, fix the hierarchy, and re-train from scratch, i.e. CoHiClust (pruning and re-train).**
> > >
> > > To address Reviewer's comment, we applied a pruning on the tree with 16 leaves and then retrain the model from scratch (with 10 leaves). The results presented in Table 7 show that such a strategy gives high similarity to ground-truth classes (acc: 0.616) but clustering metrics are slightly lower than the model with fine-tuning (acc: 0.638). Both models were trained with the same number of epochs.
> > >
> > > **Regarding the last question (Q4)**, we cannot deduce the intention of the Reviewer. It seems that part of the question has been removed. Could you ask your question once more?

---

> > > > ### Comment · Reviewer_PDcK · 2022-11-24
> > > > **Reviewer Response**
> > > >
> > > > I thank the effort made by the authors. The 4th point is not a problem, thank you for your attention.
> > > >
> > > > I'd like to encourage the author to keep improving the method and test it in large-scale datasets to have more promising results in large-scale datasets.

---

> ### Author Response · Authors · 2022-11-13
> **Authors' response to Reviewer PdCK (part 1)**
>
> Thank you for your detailed review.
>
> **Q1: The qualitative results are a bit limited for people to understand if the approach really can generate proper hierarchies. Not many ablation studies are included.**
>
> We extended the ablation study. In the appendix, we investigated the following aspects:
> * How particular components of the loss function affect the final results (app. B.2),
> * How does the model perform if we modify the loss function, i.e. eliminate the square root from the CoHiLoss (B.3),
> * What are possible ways for generating hierarchies with a smaller number of clusters (B.4).
>
> We hope that these analyses will be of interest and will help the reader to understand how our method works. We refer the Reviewer to appendix B for detailed results and discussion.
>
> **Q2: The reviewer is unsure if Mautz et al is the only related hierarchical clustering proposed in the deep learning era.**
>
> We are aware that there is vast literature concerning hierarchical classification models or approaches, which apply the shallow hierarchical clustering method to the representation taken from pre-trained models (in CV or NLP). We included some of these papers in the related works and we are ready to extend this section further. However, we did not find any hierarchical clustering models based on deep neural networks which we could directly compare with, except DeepECT. Moreover, none of the ICLR Reviewers recommend models to compare with what we believe reflects the current knowledge.
>
> To extend the experimental study on color images, we additionally applied typical agglomerative clustering. In the first case, it was applied to raw image data, while in the second case, it was evaluated on the representation taken from our contrastive model. The results in table below (or see Table 3 in the revised version) show a big gap between our approach and classical techniques even if both algorithms were run on the same representation.
>
> |                             | CIFAR-10 | STL-10 | IN-10 | IN-Dogs |
> |-----------------------------|----------|--------|-------|---------|
> | Agglom. on raw image        | 0.085    | 0.079  | 0.181 | 0.101   |
> | Agglom. on pre-trained rep. | 0.350    | 0.464  | 0.530 | 0.252   |
> | CoHiClust                   | 0.532    | 0.503  | 0.742 | 0.407

---

### Official Review · Reviewer_Xeet · 2022-10-26

**Confidence:** 3
**Correctness:** 2
**Technical Novelty And Significance:** 2
**Empirical Novelty And Significance:** 2
**Recommendation:** 5

**Clarity, Quality, Novelty And Reproducibility:**

With both hierarchical clustering and contrastive learning techniques, it seems straightforward to get the extensions of previous approaches. However, the experiments indicate that they do not work better than baseline approaches. It would have been nice if the authors had also discussed ways in which one or more techniques could be combined to deal with the lower NMI though, or identified more suitable large-scale datasets to test and justify the efficacy of the proposed model.

**Strength And Weaknesses:**

Strength: The paper is written clearly. The topic is somehow interesting.
The paper evaluates on various datasets.

Weakness:

1. The authors claim that the pruning strategy will "Assume that the importance of the cluster is related to the average number of assigned examples. We reduce a leaf with the lowest probability. After removing the leaf, we finetune the whole model using CoHiClust loss. If we want to reduce more leaves, we perform leaf pruning and model retraining iteratively until the requested number of leaves is obtained." This might just take a much longer time to re-train the model, which needs more discussion and comparison on the training efficiency.
2. CC and PIC often show better performance than the proposed CoHiClust with a large margin, which might suggest that the authors look into the deeper reason for the results and see if any suitable hierarchical clustering data can be utilized in the experiments that can justify the efficacy of the method.

**Summary Of The Paper:**

This paper proposed a contrastive hierarchical clustering model CoHiClust which attempts to learn a binary tree clustering structure rather than typical flat clustering.
A pruning strategy is introduced to create a fixed number of leaves by removing a leaf and retraining the model iteratively.
 Experimental analysis performed on typical clustering benchmarks confirms that the produced partitions have high similarity with ground truth classes.

**Summary Of The Review:**

Overall it is not difficult to parse and is well-organized. However, the paper’s claims recall more large-scale datasets in the experiment. The pruning stage takes a long time according to the tree depth, which in the worst case is very high.

---

> ### Author Response · Authors · 2022-11-11
> **Authors' response to Reviewer Xeet (part 2)**
>
> **Q3: It seems straightforward to get the extensions of previous approaches. It would have been nice if the authors had also discussed ways in which one or more techniques could be combined to deal with the lower NMI, or identified large-scale datasets to test the proposed.**
>
> First, we would like to emphasize that optimizing (supervised) clustering metrics should not be the primary goal of clustering. Data clustering is usually used to deliver information about the structure of data. From this perspective, hierarchical models are often preferable over flat clustering models.
>
> Moreover, we argue that extending arbitrary flat clustering approaches to the hierarchical case using contrastive learning is not straightforward. The following questions arise in the construction of the hierarchical clustering:
> * how to transform neural network responses to decisions in a tree?
> * how to design loss function applicable to hierarchical structures?
> * what regularizations are required in training?
> * how to cut the hierarchy to a given number of groups?
>
> In CoHiClust, we give possible answers to these questions, which could inspire the community in extending our work.
>
> Tables 1 and 2 reported in the paper show that simple hierarchical clustering models do not produce satisfactory results even on simple image data. To further investigate the performance of straightforward hierarchical approaches, we additionally applied typical agglomerative hierarchical clustering on color images (either on raw images or on the representation taken from our model). The results in the table below (as well as in Table 3 in the revised version) show a big gap between our approach and agglomerative techniques even if both algorithms were run on the same representation.
>
>
> |                             | CIFAR-10 | STL-10 | IN-10 | IN-Dogs |
> |-----------------------------|----------|--------|-------|---------|
> | Agglom. on raw image        | 0.085    | 0.079  | 0.181 | 0.101   |
> | Agglom. on pre-trained rep. | 0.350    | 0.464  | 0.530 | 0.252   |
> | CoHiClust                   | 0.532    | 0.503  | 0.742 | 0.407
>
> Taking the above issues into account, we believe that discussing possible extensions of previous works to the hierarchical case exceeds the scope of our conference paper. Our goal was to initiate the research on hierarchical clustering models in deep learning, which can be continued in the following works. To partially address the Reviewer's comment, we indicated possible future directions in the last section (Conclusion and future works) in the revised version.

---

> ### Author Response · Authors · 2022-11-13
> **Authors' response to Reviewer Xeet (part 1)**
>
> Thank you for the comments and interesting insights.
>
> **Q1: Pruning might just take a much longer time to re-train the model, which needs more discussion and comparison on the training efficiency.**
>
> Fine-tuning the model during the pruning phase allows the model slowly but surely adapt to changes in its architecture. By default, we use the same number of epochs to build the complete hierarchy as well as in the pruning phase (including fine-tuning). In consequence, pruning doubled the training time of our final model.
>
> We would like to emphasize that pruning may not be required in practice. Our model includes two regularization terms: entropy ($R_1$) and NT-XENT ($R_2$). We performed an additional ablation study, which shows that CoHiClust reduces the number of clusters automatically if we eliminate the entropy regularization from the loss. Alternatively, we can also perform pruning without fine-tuning the model (this is identical to typical pruning used in shallow decision trees). The following table shows the comparison:
>
> |                                     | NMI     | #Clusters |
> |-------------------------------------|-------|-----------|
> | CoHiClust (w/o entropy)             | 0.486  | 9         |
> | CoHiClust (pruning and fine-tuning) | 0.532  | 10        |
> | CoHiClust (pruning only)             | 0.461 | 10        |
>
> The pruning with fine-tuning (default strategy) presents the best performance, but the model without entropy regularization also produces good results. The model without fine-tuning obtains the worst performance, which shows that fine-tuning is an essential part. We added this ablation study to appendix B.4 of the revised version.
>
>
> **Q2: CC and PIC often show better performance, which might suggest that the authors look into the deeper reason for the results and see if any suitable hierarchical clustering data can be utilized in the experiments.**
>
> We do not need a specific hierarchical dataset to apply hierarchical clustering algorithms. Every dataset can be analyzed at various granularity and we can inspect the relations between groups of the resulting hierarchy. In the paper, we discuss the hierarchies for FMNIST (caption below Figure 1) and ImageNet-10 (caption below Figure 3 and the paragraph at the bottom of page 7).
>
> A major advantage of CoHiClust over flat clustering algorithms is that it brings extra information about relations between clusters. We believe that this information is beneficial for practitioners. This is confirmed by the popularity of hierarchical algorithms in bioinformatics. Typical flat clustering methods can often obtain better metrics but cannot deliver such rich information about clusters.

---

### Decision · Program_Chairs · 2023-01-20

**Decision:**

Reject

**Justification For Why Not Higher Score:**

The paper proposes a very interesting approach with an extensive quantitative methods. There are some weaknesses that should be addressed as the reviewer mentioned before being accepted although they have been addressed in the author feedback. The first issue is to evaluate the training efficiency of the proposed method that should depend on the fixed height and the baselines, and compare the training time with the performance. The second issue is the height of the method and the number of classes of the datasets, why all the experiments are conducted with datasets of 10 classes, how does it change with different number of classes. Related with the previous comment, the paper should be trained on ImageNet as a reviewer mentioned.

I encourage the author to keep working on the paper thanks to its potential, but needs some improvements before being accepted.

**Justification For Why Not Lower Score:**

N/A

**Metareview: Summary, Strengths And Weaknesses:**

# Summary
This paper focuses on hierarchical clustering with a designed contrastive objective. The proposed approach attempts to learn a binary tree clustering structure rather than typical flat clustering by decomposing the classification into a series of the decision process. Then two regularization strategies are used to make the base neural network train with the arbitrary number of leaves, and a pruning strategy is introduced to create a fixed number of leaves by removing a leaf and retraining the model iteratively. The experiments conducted on large-scale image datasets shows the CoHiClust better clustering performance compared to some existing deep and flat clustering algorithms.
# Strengths:
- The paper is written clearly and easy to follow.
- The problem is very interesting for representation learning.
- The proposed idea of using the classification as a binary tree decision process is interesting.
- Extensive results in MNIST, FMNIST, CIFAR-10, STL-10, IN-10, and IN-Dogs show the strength of the proposed hierarchical method
# Weaknesses:
The major weaknesses raised by the reviewer are the following:
- Training efficiency due to the training from scratch the model after pruning.
- The proposed method assumes fixed height, which is a major limitation.
- Current experiments are limited to ImageNet-10 and ImageNet-Dogs. Training on ImageNet and comparing standard contrastive learning results can help here.
- Issues with the novelty of proposed method that only uses some self-supervised methods for representation learning and combines them with some hierarchical clustering techniques Although there are no specific models for performing hierarchical clustering under the contrastive learning framework.



**Summary Of Ac-Reviewer Meeting:**

N/A